# It's how you say it: Systematic A/B testing of digital messaging cut hospital no-show rates

**Adi Berliner Senderey**[1,2☯]*, **Tamar Kornitzer**[3], **Gabriella Lawrence**[1,4], **Hilla Zysman**[3], **Yael Hallak**[5], **Dan Ariely**[3,5☯], **Ran Balicer**[1,6☯]

**1** Clalit Research Institute, Clalit Health Services, Tel Aviv, Israel, **2** The Faculty of Industrial Engineering and Management, Technion–Israel Institute of Technology, Haifa, Israel, **3** Kayma Labs, kayma, Tel Aviv, Israel, **4** Braun School of Public Health, Hebrew University–Hadassah Medical Center, Jerusalem, Israel, **5** Fuqua School of Business, Duke University, Durham, North Carolina, United States of America, **6** Public Health Department, Ben Gurion University of the Negev, Be'er Sheva, Israel

☯ These authors contributed equally to this work.
* adiber3@gmail.com

**Data Availability Statement:** The data underlying the results presented in the study are available from clalit health services http://clalitresearch.org/.

## Abstract

Failure to attend hospital appointments has a detrimental impact on care quality. Documented efforts to address this challenge have only modestly decreased no-show rates. Behavioral economics theory has suggested that more effective messages may lead to increased responsiveness. In complex, real-world settings, it has proven difficult to predict the optimal message composition. In this study, we aimed to systematically compare the effects of several pre-appointment message formats on no-show rates. We randomly assigned members from Clalit Health Services (CHS), the largest payer-provider healthcare organization in Israel, who had scheduled outpatient clinic appointments in 14 CHS hospitals, to one of nine groups. Each individual received a pre-appointment SMS text reminder five days before the appointment, which differed by group. No-show and advanced cancellation rates were compared between the eight alternative messages, with the previously used generic message serving as the control. There were 161,587 CHS members who received pre-appointment reminder messages who were included in this study. Five message frames significantly differed from the control group. Members who received a reminder designed to evoke emotional guilt had a no-show rates of 14.2%, compared with 21.1% in the control group (odds ratio [OR]: 0.69, 95% confidence interval [CI]: 0.67, 0.76), and an advanced cancellation rate of 26.3% compared with 17.2% in the control group (OR: 1.2, 95% CI: 1.19, 1.21). Four additional reminder formats demonstrated significantly improved impact on no-show rates, compared to the control, though not as effective as the best performing message format. Carefully selecting the narrative of pre-appointment SMS reminders can lead to a marked decrease in no-show rates. The process of a/b testing, selecting, and adopting optimal messages is a practical example of implementing the learning healthcare system paradigm, which could prevent up to one-third of the 352,000 annually unattended appointments in Israel.

**Funding:** The Israeli Ministry of Finance provided funds to the commercial company "Kayma Labs" to permit the development of the randomization process. No other funds were received by the authors in connection with this study. No funding bodies had any role in the study design, data collection, analysis, decision to publish, or preparation of the manuscript. The authors TK, HZ, and DA are employed by "Kayma Labs", though this company did not fund this research paper.

**Competing interests:** The authors TK, HZ, and DA are employed by the commercial company "Kayma Labs". This commercial affiliation does not alter our adherence to PLOS ONE policies on sharing data and materials.

## Introduction

Unattended medical appointments are a frequent event. Hospital outpatient clinics have reported no-show rates of 19.3%-43.0% globally [1]. These events negatively impact care quality worldwide, causing major disruptions to clinical management, delays in scheduled care, and reduced patient contentedness. Health-care providers consider the no-show phenomenon as intractable and invest extensive efforts and resources to control and create workarounds such as overbooking [1–4].

Short message services (SMSs) are frequently used as pre-appointment reminders by health service providers to reduce appointment no-shows, and to provide information that relates to, and encourages, the desired behavior of canceling or keeping the appointment [5–11]. There is strong evidence that even simple SMS reminders are effective in reducing non-attendance compared to no reminders at all, though their impact is small [3, 12–14]; thus, the reminders' performance is considered sub-optimal. Health providers who use reminders still experience substantial no-show rates of 21%-25%, resulting in decreased quality of care [3, 9, 12, 13, 15, 16]. Simple straightforward reminders implicitly assume that one key reason a patient does not attend their appointment is due to forgetfulness. Yet, there is vast evidence suggesting that other reasons for non-attendance without notification are more prominent and that a more holistic approach to this issue is needed [17–20].

Previous studies have demonstrated that the strategic narrative of the reminder may increase compliance in the healthcare domain. For example, a study that focused on human papillomavirus (HPV) infection investigated the impact of differential text messages on child HPV vaccination rates and found that persuasive text reminders, emphasizing the potential threat for the child, improved HPV vaccination rates [21].

Additionally, findings from two randomized control trials concluded that missed hospital appointments might be reduced by rephrasing appointment reminders and stating appointment costs [3]. The specific cost of the appointment manipulated guilt emotions, which in turn led to the lowest no-show rates. However, mentioning specific costs can antagonize or make patients doubt the authenticity of the reminder. We suggest that the same effect of guilt can be evoked without stating the cost, by emphasizing emotional guilt. In addition, there is no consensus as to the motivators that would optimally increase the likelihood that members will either attend their appointment or cancel it in advance.

Behavioral economics theory suggests that different motivational narratives, such as fairness to others or adherence to social norms, can dramatically increase a message's impact as compared with a generic informative format [21–28]. Many retail and finance industries' policies and practices reflect these theories and are designed to prompt people towards particular choices. Such policies and practices might have been relatively underused in healthcare practice.

This study aimed to assess whether using specific message formats for appointment reminders influences advanced cancellation and no-show rates. We examined whether a change in the narrative of the current SMS reminder increased members' engagement compared to the SMS reminder currently in use, and whether specific strategic narratives are more effective than others. Implementing our research results in daily clinical practice can affect members' behavior, improve quality of care, and potentially serve as a practical example for the learning health system [29].

## Materials and methods

### Setting and data sources

This study was based on data from individuals with scheduled appointments to one of 596 outpatient clinics located within 14 Clalit Health Services (CHS) hospitals. CHS is the largest

payer-provider healthcare organization in Israel, which provides primary, specialty, and inpatient care to over 52% of the Israeli population and has 4.4 million members. CHS's comprehensive healthcare data warehouse combines hospital and community medical records. All Israeli citizens are covered by one of four healthcare organizations, and while it is possible to switch between the organizations, membership turnover within CHS is less than 1% annually [30], allowing for consistent longitudinal follow-up. CHS' electronic health records (EHR) contain administrative and clinical data, socio-demographic information, diagnoses from community and hospital settings, recorded chronic diseases, clinical markers, and appointment related details. All CHS members' information was extracted from CHS's EHR, as of the index date (appointment date) and from current demographics.

CHS operates an appointment reminder system that automatically sends a text message five days prior to a scheduled appointment with a link to an internet-based system that allows for confirmation or cancellation the appointment in advance. Data from the CHS SMS appointment reminder system was retrieved and appended to the abovementioned data points.

## Study population and design

The population eligible for this study included all CHS members who were 18 years old and older, with scheduled appointments between December 1, 2018 to March 31, 2019, at one of the 596 outpatient clinics within CHS' 14 hospitals. All participants had a valid cell phone number in the CHS EHR and consented to receive phone-based appointment reminders. The index date was defined as the date of the scheduled appointment (Fig 1). Randomization occurred via a randomization program and participants were assigned to one out of nine messages that were issued five days in advance of the appointment. Members with multiple appointments during the study period could receive the same or differently framed messages for each appointment.

## Variables definitions

**Appointment reminders.** CHS members were randomly assigned to one out of nine possible message frames that reminded them of their upcoming appointment (see Table 1). Eight variations were designed based on the following principles: the 'social norm' versions

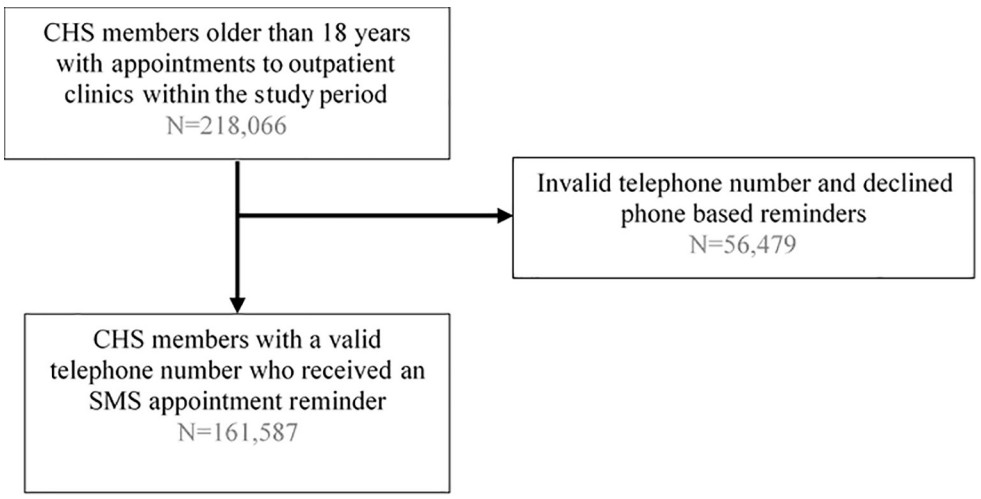

**Fig 1. Flow chart of the population.**

**Table 1. Different framings of SMS reminders sent to members five days prior to the scheduled appointment.**

| | |
|---|---|
| **Control message** | Hello, this is a reminder for a hospital appointment you have scheduled. Click the link to confirm or cancel attendance to the appointment |
| **Standard message** | Hello, you have a hospital appointment at [clinic] on [date] at [time]. Click the link to confirm or cancel attendance to the appointment |
| **Personal request message** | Hello, this is Lior from [name] hospital. I wanted to remind you of the appointment you scheduled. Click the link to confirm or cancel attendance to the appointment |
| **Professional figure message** | Hello, your caregiving physician wishes to remind you that you have scheduled an appointment and looks forward to seeing you in the clinic. Click the link to confirm or cancel attendance to the appointment |
| **Appointment cost message** | Hello, this is a reminder for a hospital appointment you have scheduled. Non-attendance without advanced notice costs National Health Services approximately 200 NIS*. Click the link to confirm or cancel attendance to the appointment |
| **Emotional relatives message** | Hello, this is a reminder for a hospital appointment you have scheduled. Your family members will be pleased to know that you are taking care of your physical state. Click the link to confirm or cancel attendance to the appointment |
| **Emotional guilt message** | Hello, this is a reminder for a hospital appointment you have scheduled. Not showing up to your appointment without canceling in advance delays hospital treatment for those who need medical aid. Click the link to confirm or cancel attendance to the appointment |
| **Social norm message** | Hello. Join the national effort to shorten appointment availability, and let us know if you intend to attend the appointment you have scheduled. Click the link to confirm or cancel attendance to the appointment |
| **Social identity message** | Hello, this is a reminder for a hospital appointment you have scheduled. Most of the patients in our clinic make sure to confirm their appointment in advance. Click the link to confirm or cancel attendance to the appointment |
| **Emotional relatives message** | Hello, this is a reminder for a hospital appointment you have scheduled. Your family members will be pleased to know that you are taking care of your physical state. Click the link to confirm or cancel attendance to the appointment |
| **Emotional guilt message** | Hello, this is a reminder for a hospital appointment you have scheduled. Not showing up to your appointment without canceling in advance delays hospital treatment for those who need medical aid. Click the link to confirm or cancel attendance to the appointment |

Abbreviations: NIS, New Israeli Shekel

*200 New Israeli Shekels equal approximately 55.55 US Dollars or 49.43 Euros

These messages were translated from the original Hebrew version.

highlighted the idea that social identity and descriptive norms potentially motivate individuals to perform certain actions [31–34]. The 'emotional' versions aimed to provoke an emotional reaction in order to prompt members to take action by mentioning people they care about [35] or to evoke feelings of sympathy or empathy [36]. The 'appointment cost' version was based on the opportunity cost effect [37]. Although members do not directly pay the healthcare organization for missed appointments, this narrative highlights the amount of money they cause the organization to lose by not showing up [38]. Both 'professional figure' and 'personal' versions relied on the messenger effect that suggests that people's compliance to a message is affected by the figure who delivers it, for example, an actual name or authority figure rather than an automated device [39]. The ninth frame was the routinely used reminder message, in use by CHS in recent years, and was retained as the control group.

**Outcomes.** The primary outcome was a no-show event, defined as a scheduled appointment that a CHS member failed to attend. The no-show rates were calculated as the number of no-show appointments out of the total number of appointments scheduled. The secondary outcome was advanced cancellation, defined as members who cancel their scheduled appointment in advance of the appointment date/time. The advanced cancellation rates were calculated as the number of cancellations out of the total number of appointment reminders sent.

**Baseline measurements.**   Sociodemographic variables were measured at index date and included biological sex, age (years), socioeconomic status (SES; low, medium, high; based on clinic-level data), population sector (Jewish, non-Jewish), and immigrant status (immigrated to Israel, born in Israel). Clinical characteristics included smoking status (current, former, or non-smoker, as reported in the EHR), body mass index (computed from documented weight and height measurements), and Charlson Comorbidity Index (computed from risk factors to evaluate an age-comorbidity score [40]). Comorbidity variables were evaluated as of the index date, and included cardiovascular diseases (yes/no; defined as any of the following: acute myocardial infarction unstable angina pectoris, angina pectoris, acute coronary syndrome, percutaneous transluminal coronary angioplasty, coronary artery bypass graft, ischemic heart disease, ischemic stroke), diabetes (yes/no), chronic kidney disease (yes/no; defined as the last eGFR value prior to index date less than 60 ml/min/1.73m2), celiac disease (yes/no), and inflammatory bowel disease (yes/no) (see S1 Table). We extracted these diagnoses from community and hospital records, as well as from the CHS chronic disease registry.

The appointment characteristics included past non-attendance–i.e.,'chronic no-show' (yes/no; defined as members who missed appointments at least two times in a row within one year prior to index date), time to an appointment (calculated as the difference in days between the date of scheduling the appointment to the date of the appointment), and clicking on the link (yes/no; defined as whether member clicked on the link in the SMS reminder message).

## Statistical analysis

Socio-demographic characteristics, clinical, and appointment-related variables were calculated within the nine different SMS groups. Summaries of continuous variables are presented as means and standard deviations unless skewed, and in that case, are presented as medians and interquartile ranges. Categorical variables are presented as absolute numbers and percentages, as appropriate.

In order to assess whether any of the message frames caused a lower risk for no-shows compared to the currently existing message frame, multinomial testing was performed. Univariate and multivariate analyses using binary logistic regression models accounting for features determined via an automated generic framework to be most predictive of no-show (i.e., socio-demographic characteristics, clinical, and appointment related variables) behavior.

Message frames were considered as treatment variables, with the existing message serving as the reference group and the record of attendance as the binary outcome variable. Secondary analyses were conducted to assess the effect of the different message frames on canceling appointments in advance.

Statistical analyses were conducted using the R language (version 3.5.3, R Foundation for Statistical Computing, Vienna, Austria). All statistical tests were 2-tailed, and a 5% significance threshold was maintained.

## Ethics

This study was reviewed by the IRB of the CHS organization and it was determined that this study was not a clinical trial, but rather an organizational initiative to optimize internal policy. It received an exemption for the need for individual informed consent since it was determined that the various intervention arms posed no harm to members. It was not registered as a clinical trial for these reasons. Obtaining consent would introduce a burden to the members (larger than the intervention itself); obtaining informed consent would cause serious practical problems that would undermine the trial results (particularly for the control group), and the risk of

harm was low since the intervention merely consisted of small modifications to existing routine processes.

## Results

During the study period, between December 2018 and March 2019, there were 218,066 scheduled appointments in CHS's hospital outpatient clinics, of which 161,587 had a valid associated mobile telephone number with approval for receiving phone-based appointment reminders (Fig 1). Among those who received one of the nine SMS appointment reminders (Table 1), in 104,469 (64.6%) cases, reminder's accompanying link was opened within 48 hours of receiving the message.

Socio-demographic, clinical, and appointment related characteristics by type of appointment reminder can be seen in Table 2. Approximately half of the eligible population was female (55.4%), and the average age of the population was 59.3 years. The distribution of all members' characteristics and appointment information was similar between the nine treatment groups, and no significant differences were found (e.g., all p values > 0.05) (Table 2).

Fig 2 and Table 3 present no-show and advanced cancellation rates in the groups receiving one of the eight alternative message frames compared with the generic control. Five out of the eight alternative message reminders presented in Table 1 ('appointment cost', 'emotional relatives', 'emotional guilt', 'social norm', and 'social identity') had significantly lower rates of no-shows and higher rates of canceling in advance compared with the routinely used message reminder. The 'emotional guilt' reminder frame led to the lowest no-show and highest advanced cancellation rates. Members who received the 'emotional guilt' message reminder had a no-show rate of 14.2% compared with 21.1% in the control group (odds ratio [OR]: 0.69, 95% confidence interval [CI]: 0.67, 0.76), and an advanced cancellation rate of 26.3% compared with 17.2% in the control group (OR: 1.2, 95% CI: 1.19, 1.21).

Favorable results were found among members who received the 'appointment cost' message, with a 15.3% no-show rate (OR: 0.72, 95% CI: 0.68, 0.77) and a 27.4% advanced cancellation rate (OR: 1.25, 95% CI: 1.1, 1.27). Similar results of 15.6% (OR: 0.77, 95% CI: 0.79, 0.82) no-show rate and 23.4% (OR: 1.18, 95% CI: 1.17, 1.19) advanced cancellation rate were found among the members who received the 'emotional relatives' framed message.

Both 'social norm' and 'social identity' framed messages were associated with a 17.8% (OR: 0.73, 95% CI: 0.61, 0.79) and 17.7% (OR: 0.83, 95% CI: 0.76, 0.87) no-show rate and 21.8% (OR: 1.08, 95% CI: 1.07, 1.08) and 24.6% (OR: 1.19, 95% CI: 1.11, 1.24) advanced cancellation rate, respectively. The 'standard', 'personal request', and 'professional figure' messages did not produce significantly different results as compared with the control message (Fig 2, Table 3).

The multivariate analysis showed that the relative reduction in the risk of no-show and advanced cancellation remained quite unchanged after adjusting for socio-demographic variables, clinical and appointment-related characteristics, and past non-attendance behavior (Table 3).

## Discussion

We have shown that careful design of SMS narratives based on behavioral economic principles can reduce hospital outpatient clinic no-show rates by over 30 percent. Out of nine differently framed reminders, five produced statistically significant lower no-show rates and higher advanced cancellation rates. The emotional guilt and specific cost message frames showed the greatest nominal differences in no-show rates and advanced cancellation rates compared with the control group (14.2% and 15.3% compared to 21.1% in no-show rates and 26.3% and 27.4% compared to 17.2% in advanced cancellation rates, respectively).

**Table 2. Socio-demographic, clinical and appointment-related characteristics by SMS group.**

| Variables | Population | Control | Standard | Social norm | Social identity | Emotional relatives | Emotional guilt | Appointment cost | Personal Request | Professional figure |
|---|---|---|---|---|---|---|---|---|---|---|
| **Individuals**, N | 161,587 | 18,086 | 18,038 | 17,467 | 17,937 | 17,501 | 17,6 46 | 18,156 | 18,103 | 18,653 |
| **Female**, n (%) | 89,599 (55.4%) | 10,070 (55.7%) | 10,040 (55.7%) | 9,696 (55.5%) | 9,865 (55.0%) | 9,783 (55.9%) | 9,904 (56.1%) | 10,120 (55.7%) | 10,019 (55.3%) | 10,102 (54.2%) |
| **Age,** mean (SD) | 59.3 (18.3) | 59.5 (18.4) | 59.6 (18.2) | 59.7 (18.4) | 59.3 (18.2) | 58.8 (18.2) | 59.2 (18.3) | 59.6 (18.1) | 58.8 (18.7) | 59.2 (17.9) |
| **Socio-economic status**, n (%) | | | | | | | | | | |
| Low | 25,702 (16.1%) | 2,962 (16.6%) | 2,789 (15.6%) | 2,719 (15.7%) | 2,828 (15.9%) | 2,834 (16.4%) | 2,855 (16.3%) | 2,969 (16.5%) | 2,807 (15.7%) | 2,939 (15.9%) |
| Medium | 57,829 (36.2%) | 6,332 (35.4%) | 6,703 (37.6%) | 6,177 (35.8%) | 6,130 (34.6%) | 6,431 (37.1%) | 6,371 (36.5%) | 6,448 (35.9%) | 6,563 (36.7%) | 6,674 (36.2%) |
| High | 76,270 (47.7%) | 8,573 (48.0%) | 8,330 (46.7%) | 8,381 (48.5%) | 8,776 (49.5%) | 8,060 (46.5%) | 8,248 (47.2%) | 8,556 (47.6%) | 8,510 (47.6%) | 8,836 (47.9%) |
| Missing | 1,786 (1.1%) | 219 (1.2%) | 216 (1.2%) | 190 (1.1%) | 203 (1.1%) | 176 (1.0%) | 172 (1.0%) | 183 (1.0%) | 223 (1.2%) | 204 (1.1%) |
| **Sector**, n (%) | | | | | | | | | | |
| Non-Jewish | 14,883 (9.2%) | 1,745 (9.6%) | 1,642 (9.1%) | 1,572 (9.0%) | 1,684 (9.4%) | 1,702 (9.7%) | 1,631 (9.2%) | 1,690 (9.3%) | 1,509 (8.3%) | 1,708 (9.2%) |
| Jewish | 146,704 (90.8%) | 16,341 (90.4%) | 16,396 (90.9%) | 15,895 (91.0%) | 16,253 (90.6%) | 15,799 (90.3%) | 16,015 (90.8%) | 16,466 (90.7%) | 16,594 (91.7%) | 16,945 (90.8%) |
| **Immigrants**, n (%) | 68,082 (42.1%) | 7,660 (42.4%) | 7,438 (41.2%) | 7,600 (43.5%) | 7,429 (41.4%) | 7,253 (41.4%) | 7,525 (42.6%) | 7,609 (41.9%) | 7,580 (41.9%) | 7,988 (42.8%) |
| **Clinical characteristics** | | | | | | | | | | |
| **Smoking status**, n (%) | | | | | | | | | | |
| Non-smoker | 91,017 (61.9%) | 10,237 (62.0%) | 10,076 (61.8%) | 9,990 (62.9%) | 10,075 (61.4%) | 9,785 (61.7%) | 9,944 (62.0%) | 10,510 (63.1%) | 10,105 (61.7%) | 10,295 (60.8%) |
| former smoker | 36,849 (25.1%) | 4,106 (24.9%) | 4,154 (25.5%) | 4,007 (25.2%) | 4,168 (25.4%) | 3,933 (24.8%) | 3,923 (24.4%) | 4,014 (24.1%) | 4,202 (25.7%) | 4,342 (25.6%) |
| Current smoker | 19,120 (13.0%) | 2,166 (13.1%) | 2,075 (12.7%) | 1,894 (11.9%) | 2,153 (13.1%) | 2,140 (13.5%) | 2,181 (13.6%) | 2,141 (12.8%) | 2,063 (12.6%) | 2,307 (13.6%) |
| Missing | 14,601 (9.0%) | 1,577 (8.7%) | 1,733 (9.6%) | 1,576 (9.0%) | 1,541 (8.6%) | 1,643 (9.4%) | 1,598 (9.1%) | 1,491 (8.2%) | 1,733 (9.6%) | 1,709 (9.2%) |
| **BMI,** mean (SD) | 27.3 (5.5) | 27.3 (5.3) | 27.2 (5.5) | 27.4 (5.4) | 27.4 (5.5) | 27.3 (5.5) | 27.3 (5.3) | 27.2 (5.4) | 27.3 (5.6) | 27.3 (5.6) |
| **Charlson score,** mean (SD) | 4.2 (3.4) | 4.3 (3.5) | 4.2 (3.4) | 4.3 (3.6) | 4.1 (3.4) | 4.1 (3.4) | 4.2 (3.5) | 4.2 (3.4) | 4.2 (3.5) | 4.2 (3.4) |
| Missing, n (%) | 18,017 (11.2%) | 1,932 (10.7%) | 2,135 (11.8%) | 1,941 (11.1%) | 1,963 (10.9%) | 1,989 (11.4%) | 1,927 (10.9%) | 1,896 (10.4%) | 2,175 (12.0%) | 2,059 (11.0%) |
| **Cardiovascular diseases**, n (%) | 32,141 (19.9%) | 3,706 (20.5%) | 3,591 (19.9%) | 3,740 (21.4%) | 3,404 (19.0%) | 3,366 (19.2%) | 3,506 (19.9%) | 3,718 (20.5%) | 3,370 (18.6%) | 3,740 (20.1%) |
| **Diabetes**, n (%) | 47,804 (29.6%) | 5,233 (28.9%) | 5,340 (29.6%) | 5,432 (31.1%) | 5,391 (30.1%) | 5,087 (29.1%) | 5,145 (29.2%) | 5,438 (30.0%) | 5,144 (28.4%) | 5,594 (30.0%) |
| **CKD**, n (%) | 77,133 (51.7%) | 8,732 (52.0%) | 8,572 (51.7%) | 8,644 (53.4%) | 8,478 (51.1%) | 8,123 (50.4%) | 8,400 (51.8%) | 8,800 (52.1%) | 8,420 (50.9%) | 8,964 (52.0%) |
| **Celiac**, n (%) | 844 (0.5%) | 97 (0.5%) | 68 (0.4%) | 99 (0.6%) | 116 (0.6%) | 108 (0.6%) | 60 (0.3%) | 105 (0.6%) | 86 (0.5%) | 105 (0.6%) |
| **IBD**, n (%) | 3,379 (2.1%) | 443 (2.4%) | 408 (2.3%) | 330 (1.9%) | 383 (2.1%) | 328 (1.9%) | 361 (2.0%) | 376 (2.1%) | 378 (2.1%) | 372 (2.0%) |
| **Appointment characteristics** | | | | | | | | | | |
| **Past behavior of non-attendance**, n (%) | 1,665 (1.0%) | 169 (0.9%) | 187 (1.0%) | 217 (1.2%) | 144 (0.8%) | 197 (1.1%) | 225 (1.3%) | 155 (0.9%) | 183 (1.0%) | 188 (1.0%) |

*(Continued)*

**Table 2.** (Continued)

| Variables | Population | Control | Standard | Social norm | Social identity | Emotional relatives | Emotional guilt | Appointment cost | Personal Request | Professional figure |
|---|---|---|---|---|---|---|---|---|---|---|
| **Time to appointment,** median (IQR) (days) | 41.0 (21.0–91.0) | 42.0 (21.0–92.0) | 42.0 (21.0–91.0) | 41.0 (20.2–91.0) | 42.0 (21.0–91.0) | 40.0 (21.0–91.0) | 39.0 (20.0–91.0) | 41.0 (20.0–91.0) | 41.0 (21.0–91.0) | 41.0 (20.0–90.0) |
| **Clicked the link,** n (%) | 104469 (64.7) | 11889 (65.7) | 11736 (65.1) | 10775 (61.7) | 11416 (63.6) | 11006 (62.9) | 11525 (65.3) | 11950 (65.8) | 11848 (65.4) | 12324 (66.1) |

Abbreviations: SMS, short message service; SD, standard deviation; IQR, interquartile range; BMI, body mass index; SES, sociodemographic status; CKD, chronic kidney disease; IBD, inflammatory bowel disease.

While many health interventions approach behavioral challenges by emphasizing the need to support and prompt the individual through reminders, these results indicate that different messages can influence no-show and cancellation rates [5, 6, 12, 41–43]. These results are aligned with behavioral economic theories. However, the current data do not offer adequate support that the varied effects have resulted from those specific psychological mechanisms. Future research is needed to explore this topic.

These results highlights the potential of introducing behavioral economics principles into multiple avenues of healthcare delivery in order to improve member adherence and reduce waste in care provision.

Our study ventures beyond the published literature by including a standard alternative message alongside the affect-based alternatives, possibly indicating that the reduction in no-show rates was in fact due to a change in context and not merely in response to a simple change in wording.

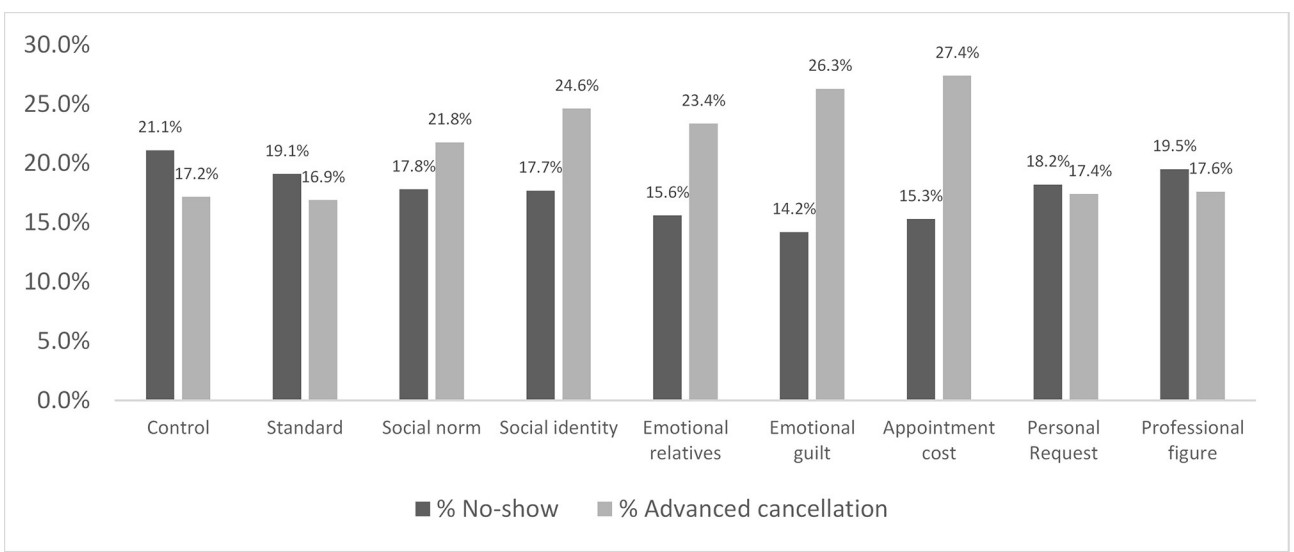

[a] Calculated as the number of people who did not attend an appointment (and did not cancel in advance) out of the total population with appointments
[b] Calculated as the number of people who canceled an appointment in advance out of the total number who clicked on the SMS link

**Fig 2. No-show[†] and advanced cancellation[‡] rates by SMS group.** Abbreviations: SMS, short message service. [†] Calculated as the number of people who did not attend an appointment (and did not cancel in advance) out of the total population with appointments. [‡] Calculated as the number of people who canceled an appointment in advance out of the total number who clicked on the SMS link.

**Table 3. The effect of the specific SMS framing on no-show or advanced cancellation according to univariate and multivariate analyses.**

| | Univariate analysis [†] | | Multivariate analysis [‡] | |
|---|---|---|---|---|
| | No-show OR [CI] | Advanced Cancellation OR [CI] | No-show OR [CI] | Advanced Cancellation OR [CI] |
| Control message | 1 | 1 | 1 | 1 |
| Standard message | 1.03 [0.97, 1.1] | 0.98 [0.94, 1.03] | 1.01 [0.96,1.1] | 0.97 [0.92, 1.03] |
| Personal request message | 0.93 [0.77, 1.23] | 1.03 (0.98,1.08] | 0.93 [0.75,1.23] | 1.02 [0.97,1.1] |
| Professional figure message | 0.87 [0.76,1.45] | 1.05 [1.0, 1.09] | 0.87 [0.78,1.46] | 1.05 [1.0, 1.1] |
| Appointment cost message | 0.72 [0.68, 0.77]*** | 1.25 [1.1,1.27]* | 0.72 [0.67,0.77]** | 1.27 [1.1,1.29]** |
| Emotional relatives message | 0.77 [0.79, 0.82]*** | 1.18 [1.17,1.19]*** | 0.77 [0.79,0.82]*** | 1.17 [1.16,1.19]** |
| Emotional guilt message | 0.69 [0.67, 0.76]*** | 1.2 [1.19,1.21]*** | 0.69 [0.67,0.75]*** | 1.2 [1.19,1.22]*** |
| Social norm message | 0.73 [0.61, 0.79]** | 1.08 [1.07,1.08]* | 0.73 [0.64,0.79]** | 1.09 [1,07,1.1]** |
| Social identity message | 0.83 [0.76, 0.87]** | 1.19 [1.11, 1.24]** | 0.82 [0.75,0.88]** | 1.19 [1.10, 1.24]** |

Abbreviations: SMS, short message service; OR, odds ratio; CI, confidence interval

[†]All analyses were based on logistic regressions

[‡] Multivariate analysis models were adjusted for age (years), sex (male or female), socioeconomic status (low or medium/high), population sector (Jewish or Non-Jewish), immigrant status (immigrated to Israel or born in Israel), smoking status (current smoker or nonsmoker/former smoker), body mass index (kg/m2), Charlson score, diagnosis of heart condition (yes or no), diagnosis of diabetes (yes or no), diagnosis of chronic kidney disease (yes or no), diagnosis of celiac (yes or no), diagnosis of inflammatory bowel disease (yes or no), past behavior of non-attendance ($\geq$2 missed appointments), time to appointment, clinic's specialty and clicked the link (yes or no).

This study had several limitations. First, all reminder messages were sent five days prior to the appointment. This five day period might be considered as relatively long, as previous studies have reported shorter periods of one to three days between sending the reminder and appointment date [6]. Sending the reminder SMS five days before the appointment may have allowed the psychological effect of the remainder to decay over time, meaning that even though the members may have confirmed attendance, they may not attend the appointment. However, if at the time of the reminder, the members had forgotten about the appointment, it may have been possible that this five day period of time will enable them to rearrange schedule in order to attend.

Another possible limitation was the inability to distinguish between members who read the SMS reminders and those who did not. However, we retain our confidence in the overall conclusions since 64.6% of the receipients clickedon the link within 48 hours, indicating that the majority of people read the message. Also, the assignment to the SMS frame for each appointment was properly randomized, and there is no reason to suspect additional potential confounders. Furthermore, all members within the study period had no more than three appointments. It is possible that receiving the same message when scheduling new appointments diminished the effect over time. The multiple and varying number of appointments per patient in the sample led to correlated error variance in the study dataset, which may have produced biased error estimates. However, due to the short period of the study (December 2018 – March 2019), not many patients had repeated appointments (approximately 9.4% of the total population).

It is important to note that the expected effect of rephrasing reminder messages may be limited, as it was designed with the 'average' person in mind, rather than customized on an individual level. While we found that the effect of sending alternative messages was maintained after adjustment by various covariates, it is possible that interaction of individual characteristics may modify the impact of specific message frames on no-shows. Future research should focus on customizing the content per person to even further reduce no-show rates.

The major strength of our findings was that all 14 CHS hospitals, located throughout the country, were included in this study. This means that the effect of different SMS versions on no-shows was assessed among participants who came from diverse backgrounds, and thus, can be generalized to a greater scale for policy implications.

The number of unattended appointments across all outpatient clinics in CHS' 14 hospitals is approximately 600,000 annually (18.7% of all outpatient clinic appointments). Our results indicate that replacing the current reminder message with a carefully designed message can potentially save 187,000 appointments annually. Nationally, this change can potentially result in saving approximately 350,000 unattended appointments, thereby improving the quality of care across the country.

Since May 2019, CHS changed the policy to adopt the "emotional guilt' narrative in all outpatient clinics for all messages used in daily practice (more than 3 million appointments a year), and is monitoring the scale of the real-world impact of this change. This can serve as an example of how research knowledge gained by a learning healthcare system can be implemented into routine clinical practice and effect changes in the organization's policy [29, 44]. It is worth noting that such a change may have additional unintended consequences other than improving visit attendance, such as a reduction in clinic or physician satisfaction ratings.

The era of digital health enables healthcare providers to systematically customize their interaction with members in order to increase the effectiveness of healthier behavior [45]. This simple example of how strategic use of traditional messaging substantially impacts members' behaviors shows the untapped potential of smart messaging in health care. Improvement in member engagement depends on the utilization of technical add-ons, but even more so, on the nuances and characterization of the way the messaging is constructed.

## Supporting information

**S1 Table. Codes used for variable definitions.**
(DOCX)

## Acknowledgments

We thank Oran Huberman, Meirav Visel, Tzahi Israel, and Moshe Gerlitz for contribution in the intervention conceptual framing and design; Sydney Krispin and Becca Feldman for their editorial support and reviewing the manuscript; and Galit Benbenishiti, Nachum Yosef, and Ilan Gofer for their help in the design and implementation of the randomization process.

## Author Contributions

**Conceptualization:** Adi Berliner Senderey, Hilla Zysman, Yael Hallak, Dan Ariely, Ran Balicer.

**Formal analysis:** Adi Berliner Senderey.

**Investigation:** Dan Ariely.

**Methodology:** Adi Berliner Senderey, Gabriella Lawrence, Dan Ariely.

**Project administration:** Adi Berliner Senderey.

**Resources:** Adi Berliner Senderey.

**Supervision:** Adi Berliner Senderey, Dan Ariely, Ran Balicer.

**Validation:** Adi Berliner Senderey.

**Visualization:** Adi Berliner Senderey, Ran Balicer.

**Writing – original draft:** Adi Berliner Senderey, Ran Balicer.

**Writing – review & editing:** Adi Berliner Senderey, Tamar Kornitzer, Gabriella Lawrence, Hilla Zysman, Dan Ariely, Ran Balicer.

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
