## [Decision Letter · Decision Letter 0]

8 May 2020

PONE-D-20-04898

It’s how you say it: systematic A/B testing of digital messaging cut hospital no-show rates

PLOS ONE

Dear Mrs. Berliner Senderey,

Thank you for submitting your manuscript to PLOS ONE. After careful consideration, we feel that it has merit but does not fully meet PLOS ONE’s publication criteria as it currently stands. Therefore, we invite you to submit a revised version of the manuscript that addresses the points raised during the review process.

We would appreciate receiving your revised manuscript by Jun 21 2020 11:59PM. To enhance the reproducibility of your results, we recommend that if applicable you deposit your laboratory protocols in protocols.io, where a protocol can be assigned its own identifier (DOI) such that it can be cited independently in the future. For instructions see: http://journals.plos.org/plosone/s/submission-guidelines#loc-laboratory-protocols

We look forward to receiving your revised manuscript.

Kind regards,

Sreeram V. Ramagopalan

Academic Editor

PLOS ONE

Journal Requirements:

We note that one or more of the authors are employed by a commercial company: Kayma labs

3. Your ethics statement must appear in the Methods section of your manuscript. If your ethics statement is written in any section besides the Methods, please move it to the Methods section and delete it from any other section. Please also ensure that your ethics statement is included in your manuscript, as the ethics section of your online submission will not be published alongside your manuscript.

Reviewers' comments:

Reviewer's Responses to Questions

**Comments to the Author**

1. Is the manuscript technically sound, and do the data support the conclusions?

Reviewer #1: Partly

2. Has the statistical analysis been performed appropriately and rigorously? 

Reviewer #1: Yes

3. Have the authors made all data underlying the findings in their manuscript fully available?

Reviewer #1: Yes

4. Is the manuscript presented in an intelligible fashion and written in standard English?

Reviewer #1: Yes

5. Review Comments to the Author

Reviewer #1: The submitted manuscript has several strong features making it worthy of publication. The topic chosen has substantial impact on both outcomes and financing in Israel’s healthcare system, and likely in all national healthcare systems. The study population and dataset have high ecological validity, the intervention design seems thorough and well-controlled, and the analyses are appropriate for the general study objectives. Most of the line-specific comments below have minor or moderate impacts on the overall quality of the study design and analysis.

The major challenge with the manuscript, as written, is that the conclusions extend well beyond what can be supported by a conservative interpretation of the data and analyses. The analyses show that varying the message content of appointment reminders can influence no-show and cancellation rates. The specific message alternatives drafted by the study authors are inspired by well-regarded and influential behavioral theories. However, the authors have offered no analyses that demonstrate whether the specific stimuli they drafted for this study are actually perceived by study participants (or people like the study participants) to have the intended affective and motivational impacts. The existing data support that different messages are more effective, but do not currently support hypotheses about _why_ they are effective.

I hate to be the reviewer that requests more data and/or analysis, especially for an open access journal submission. However, there is a straightforward fix if you want to go beyond the conclusion that “different messages can influence no-show and cancellation rates”. You’d need to get independent ratings of the message content, separately from their effect on no-show and cancellation rates. Even post hoc, even with a convenience sample of Israelis, consistent Likert-scale ratings of the messages against statements like “This message makes me feel guilt” or “This message makes me feel peer pressure” could support the argument that the stimuli you drafted are good representations of the behavioral economic principles they’re supposed to represent. I can’t tell whether any of the authors are behavioral or social psychologists, but a relatively quick consultation with a research psychologist could yield a good set of rating questions to support the various source theories.

If the authors do not wish to conduct additional analysis to show that the stimuli are rated as having the effect intended by these various theories, then a resubmitted manuscript should substantially scale back the interpretation of results. It would be appropriate, consistent with the often theory-free discipline of A/B testing, to note simply that different messages influence appointment behaviors, and imply that continued A/B testing could optimize appointment behaviors even if no underlying theory is applied. It would even be appropriate to speculate that the varied appointment effects are consistent with the source behavior theories, with a heavy “more research is needed” caveat as a limitation. But the current data do not offer adequate support that any of these behavioral theories is responsible for study effects, nor that one theory predicts stronger effects than another.

Line-specific comments:

• Abstract line 11 - appears to contain a typo “Clalit’sto”. Additional typos and grammar notes appear in subsequent lines, but my review was not exhaustive. Recommend that the resubmitted manuscript first get a thorough proofread by a native English speaker.

• Text line 53 - “randomized” not “randomize”

• Text line 61 - “suggests” not “suggest”

• Text line 61 – “Behavioral economic theory” does correlate message contents and their “different motivational narratives” with varying impact, but “_testing_ different motivational narratives” is just a good practice encouraged in behavioral economics. If I’m understanding your sentence correctly, I think it is stronger with the word “testing” removed.

• Text line 65 – Be careful making a value assertion here about “underuse” - how much behavioral economics use would be enough? How would we determine this?

• Text line 72 - “affect” not “effect”

• Text line 73 - “the learning health system” not “health learning system”

• Text line 79 - “registerd” is misspelled

• Text line 155 - The methods are not clearly described, but much of the source data for CCI appear to be diagnoses available in the system across each subject’s entire medical history with Clalit. Access to diagnoses is likely unequal across the cohort, based on patient age and duration of using this particular clinic system. This is a source of noise that may have weakened the likelihood of showing positive correlations with CCI in study results. Randomization would have balanced out the degree of “availability” bias that this would have introduced in CCI calculations between treatment arms, but the chance of detecting a CCI main effect would still be weakened throughout the population.

• Text line 191 – It appears that this is multinomial testing, not serial A/B testing as discussed in marketing disciplines. The medical/health services research audience for your paper will not necessarily know what A/B testing is, so it should be defined more explicitly, or the market research term should be removed from this article.

• Text line 209 - Text lines 301-302, later in the paper, clarify that the 64.6% statistic _only_ refers to click-through behavior. This line implies that it refers both to reading and to click-through. The number of messages that were read is likely larger than 64.6%, but is not measured or reported separately. Please clarify the language accordingly.

• Text line 215 - Table 2 contains no statistical inference tests to support that “no significant differences were found”. It is not necessary to modify Table 2 to include tests if they were done and indeed not significant. If tests were done, a parenthetical phrase in this sentence would suffice (e.g., all p values > X). If tests were not done, please remove the “no significant differences were found” phrase.

• Text line 272 - your analysis models do not support your ability to conclude that specific messages led to the “lowest” no-show rates or the “highest” advanced cancellation rates. This would require pairwise comparisons among the messages that reduced no-shows and increased cancellations. You can observe that specific messages had the greatest “nominal” differences from the control group, which acknowledges that “lowest” and “highest” are not supported by statistical inference testing. In reality, the pairwise ORs for no-shows and cancellations are quite comparable in magnitude among the five of your messages with significant differences from the control group.

• Text line 279 - “substancial“ is misspelled

• Text lines 278-281 - is your effect due to a “mere change in narrative” or specific “behavioral economics principles”? Your analyses don’t allow you to distinguish. The messages included in Supplement 1 would appear to a prudent layperson to match the principles listed in each row. But you’ve reported no independent message testing to confirm that the messages are perceived that way. This does not allow you to defend against any number of alternative explanations that have nothing to do with the specific behavioral economics principles. For example, is the difference because the effective messages have the longest word/character counts? Is this a Hawthorne effect, given that the control message had been in use for a while and most of the experimental messages were noticeable changes from the prior one? Is it because the effective messages provide a reason, any reason, to show up or cancel in advance (see Ellen Langer’s 1978 study on “placebic” information in persuasion)?

• Text lines 285-286 - You’ve shown no data to confirm whether patients receiving these messages felt any ‘affect’ or any specific emotional response, so the current version of your study cannot be evaluated against specific frameworks such as MINDSPACE.

• Text lines 306-308 - This is a minor point given the larger context of the study, but the multiple and varying number of appointments per patient in the sample leads to correlated error variance in your study dataset, which may have produced biased error estimates. If a study dataset allows some patients to count once, but others to count multiple times, a statistical analysis model that accounts for this correlation in error variances is often used. In your resubmission, it’s probably better to mention this as a limitation than to go back and apply a more sophisticated analysis model (e.g., genereralized estimating equations) that may not improve your precision.

• Text lines 327-328 - it might make sense to acknowledge that a change like this might have additional unintended consequences besides improving visit attendance. For example, does using a guilt-based message encourage patient resentment that might reduce clinic or physician satisfaction ratings?

• Text line 332 - “customize” instead of “costumize”?

6. PLOS authors have the option to publish the peer review history of their article (what does this mean?). If published, this will include your full peer review and any attached files.

Reviewer #1: Yes: Vernon F. Schabert, Ph.D.

---

## [Author Response · Author response to Decision Letter 0]

1 Jun 2020

Response: Thank you. We have reviewed PLOS ONE’s style requirements and believe we have make the correct formatting changes.

We note that one or more of the authors are employed by a commercial company: Kayma labs

a. Please provide an amended Funding Statement declaring this commercial affiliation, as well as a statement regarding the Role of Funders in your study. If the funding organization did not play a role in the study design, data collection, and analysis, decision to publish, or preparation of the manuscript and only provided financial support in the form of authors' salaries and/or research materials, please review your statements relating to the author contributions, and ensure you have specifically and accurately indicated the role(s) that these authors had in your study. You can update author roles in the Author Contributions section of the online submission form.

“The funder provided support in the form of salaries for authors [insert relevant initials] but did not have any additional role in the study design, data collection, and analysis, decision to publish, or preparation of the manuscript. The specific roles of these authors are articulated in the ‘author contributions’ section.”

Response: We have included the following section at the end of the revised manuscript: “The authors TK, HZ, DA are employed by the commercial company “Kayma Labs”, though this company did not fund this research paper. 

The Israeli Ministry of Finance funded this research but did not play a role in the study design, data collection, and analysis, decision to publish, or preparation of the manuscript and only provided financial support in the form of authors' salaries or research materials”.

Response: We have included the following section in the revised manuscript: “Competing Interests Statement: The authors TK, HZ, DA are employed by the commercial company “Kayma Labs”. This commercial affiliation does not alter our adherence to PLOS ONE policies on sharing data and materials.”

Response: We have included both an updated Funding Statement and Competing Interests Statement in our cover letter above.

3. Your ethics statement must appear in the Methods section of your manuscript. If your ethics statement is written in any section besides the Methods, please move it to the Methods section and delete it from any other section. Please also ensure that your ethics statement is included in your manuscript, as the ethics section of your online submission will not be published alongside your manuscript.

Response: The ethics statement was moved to the methods section. 

Reviewers' comments:

Reviewer's Responses to Questions

1. Is the manuscript technically sound, and do the data support the conclusions?

Reviewer #1: Partly

Response: We appreciate the reviewer’s remark and hope that the responses below (specifically # 5) will address this appropriately and help to draw better conclusions based on the data presented. 

2. Has the statistical analysis been performed appropriately and rigorously?

Reviewer #1: Yes

3. Have the authors made all data underlying the findings in their manuscript fully available?

The PLOS Data policy requires authors to make all data underlying the findings described in their manuscript fully available without restriction, with rare exceptions (please refer to the Data Availability Statement in the manuscript PDF file). The data should be provided as part of the manuscript or its supporting information, or deposited to a public repository. For example, in addition to summary statistics, the data points behind means, medians, and variance measures should be available. If there are restrictions on publicly sharing data—e.g. participant privacy or use of data from a third party—those must be specified.

Reviewer #1: Yes

4. Is the manuscript presented in an intelligible fashion and written in standard English?

Reviewer #1: Yes

5. Review Comments to the Author

Reviewer #1: The submitted manuscript has several strong features making it worthy of publication. The topic chosen has substantial impact on both outcomes and financing in Israel’s healthcare system, and likely in all national healthcare systems. The study population and dataset have high ecological validity, the intervention design seems thorough and well-controlled, and the analyses are appropriate for the general study objectives. Most of the line-specific comments below have minor or moderate impacts on the overall quality of the study design and analysis.

The major challenge with the manuscript, as written, is that the conclusions extend well beyond what can be supported by a conservative interpretation of the data and analyses. The analyses show that varying the message content of appointment reminders can influence no-show and cancellation rates. The specific message alternatives drafted by the study authors are inspired by well-regarded and influential behavioral theories. However, the authors have offered no analyses that demonstrate whether the specific stimuli they drafted for this study are actually perceived by study participants (or people like the study participants) to have the intended affective and motivational impacts. The existing data support that different messages are more effective, but do not currently support hypotheses about _why_ they are effective.

I hate to be the reviewer that requests more data and/or analysis, especially for an open access journal submission. However, there is a straightforward fix if you want to go beyond the conclusion that “different messages can influence no-show and cancellation rates”. You’d need to get independent ratings of the message content, separately from their effect on no-show and cancellation rates. Even post hoc, even with a convenience sample of Israelis, consistent Likert-scale ratings of the messages against statements like “This message makes me feel guilt” or “This message makes me feel peer pressure” could support the argument that the stimuli you drafted are good representations of the behavioral economic principles they’re supposed to represent. I can’t tell whether any of the authors are behavioral or social psychologists, but a relatively quick consultation with a research psychologist could yield a good set of rating questions to support the various source theories.

If the authors do not wish to conduct additional analysis to show that the stimuli are rated as having the effect intended by these various theories, then a resubmitted manuscript should substantially scale back the interpretation of results. It would be appropriate, consistent with the often theory-free discipline of A/B testing, to note simply that different messages influence appointment behaviors, and imply that continued A/B testing could optimize appointment behaviors even if no underlying theory is applied. It would even be appropriate to speculate that the varied appointment effects are consistent with the source behavior theories, with a heavy “more research is needed” caveat as a limitation. But the current data do not offer adequate support that any of these behavioral theories is responsible for study effects, nor that one theory predicts stronger effects than another.

Response: Thank you for this important and insightful comment. 

We agree that the existing data can only support that different messages are more effective even if no underlying theory is applied and that the current data do not support hypotheses about why they are effective or that one theory predicts stronger effects over another.

We rephrased and scaled back the interpretation of the results in our discussion section, suggesting that future research is needed in order to demonstrate whether the specific wording perceived by study participants had the intended affective and motivational impacts.

Please also see our response below to the comment regarding lines 278-281.

 

Line-specific comments:

• Abstract line 11 - appears to contain a typo “Clalit’sto”. Additional typos and grammar notes appear in subsequent lines, but my review was not exhaustive. Recommend that the resubmitted manuscript first get a thorough proofread by a native English speaker.

Response: We thank the reviewer for this comment. We corrected the typos and grammar accordingly and had a professional native English speaker proofread the revised manuscript.

• Text line 53 - “randomized” not “randomize”

Response: Corrected

• Text line 61 - “suggests” not “suggest”

Response: Corrected

• Text line 61 – “Behavioral economic theory” does correlate message contents and their “different motivational narratives” with varying impact, but “_testing_ different motivational narratives” is just a good practice encouraged in behavioral economics. If I’m understanding your sentence correctly, I think it is stronger with the word “testing” removed.

Response: We fully agree with the reviewer's comment and removed “testing” from the sentence. 

• Text line 65 – Be careful making a value assertion here about “underuse” - how much behavioral economics use would be enough? How would we determine this?

Response: We thank the reviewer for the comment. The sentence was rephrased in the manuscript accordingly: “Such policies and practices might have been relatively underused in healthcare practice”.

• Text line 72 - “affect” not “effect”

Response: Corrected

• Text line 73 - “the learning health system” not “health learning system”

Response: Corrected

• Text line 79 - “registerd” is misspelled

Response: Corrected

• Text line 155 - The methods are not clearly described, but much of the source data for CCI appear to be diagnoses available in the system across each subject’s entire medical history with Clalit. Access to diagnoses is likely unequal across the cohort, based on patient age and duration of using this particular clinic system. This is a source of noise that may have weakened the likelihood of showing positive correlations with CCI in study results. Randomization would have balanced out the degree of “availability” bias that this would have introduced in CCI calculations between treatment arms, but the chance of detecting a CCI main effect would still be weakened throughout the population.

Response: We thank the reviewer for this comment. It is important to emphasize that the study was performed using electronic health records data from Clalit Health Services (CHS), the largest of four national health funds in Israel. All Israeli citizens are covered by one of the health funds and can switch between them at any time. However, switching rates are relatively low (about 1% annually), which allows consistent longitudinal follow-up. All participants in this study had more than one-year membership at CHS, thus it is unlikely that diagnoses are missing or lacking documentation. We agree that the randomization assigning each participant to one of nine messages, balancing the biases but might weaken the power of CCI’s main effect. 

We have added the following section to the methods: “All Israeli citizens are covered by one of four healthcare organizations, and while it is possible to switch between the organizations, membership turnover within CHS is less than 1% annually, allowing for consistent longitudinal follow-up.”

• Text line 191 – It appears that this is multinomial testing, not serial A/B testing as discussed in marketing disciplines. The medical/health services research audience for your paper will not necessarily know what A/B testing is, so it should be defined more explicitly, or the market research term should be removed from this article. 

Response: We agree that the testing is not a serial A/B testing as known in marketing disciplines. We randomized each patient to one of nine message reminders and compared between the groups at a specific time point. The sentence was rephrased in the manuscript accordingly: “compared to the currently existing message frame, we performed multinomial testing”.

• Text line 209 - Text lines 301-302, later in the paper, clarify that the 64.6% statistic _only_ refers to click-through behavior. This line implies that it refers both to reading and to click-through. The number of messages that were read is likely larger than 64.6%, but is not measured or reported separately. Please clarify the language accordingly.

Response: We thank the reviewer for this comment. Indeed, due to privacy policy limitations, we could only count the click-through rate and could not track the rate at which participants opened the message. Thus, as mentioned in the comment, it is more than likely that the reading message rate was higher than 64.4%. The sentence was corrected accordingly in the manuscript: “Among those who received one of the nine SMS appointment reminders (Table 1), in 104,469 (64.6%) cases, the reminder’s’ accompanying link was opened within 48 hours of receiving the message.”

• Text line 215 - Table 2 contains no statistical inference tests to support that “no significant differences were found”. It is not necessary to modify Table 2 to include tests if they were done and indeed not significant. If tests were done, a parenthetical phrase in this sentence would suffice (e.g., all p values > X). If tests were not done, please remove the “no significant differences were found” phrase.

Response: Thank you for this important comment. All statistical analyses were conducted to find differences within the groups. We added the phrase you suggested to the manuscript: “(e.g., all p values > 0.05)”

• Text line 272 - your analysis models do not support your ability to conclude that specific messages led to the “lowest” no-show rates or the “highest” advanced cancellation rates. This would require pairwise comparisons among the messages that reduced no-shows and increased cancellations. You can observe that specific messages had the greatest “nominal” differences from the control group, which acknowledges that “lowest” and “highest” are not supported by statistical inference testing. In reality, the pairwise ORs for no-shows and cancellations are quite comparable in magnitude among the five of your messages with significant differences from the control group.

Response: We thank the reviewer for this insightful comment. It is agreed that we can only compare each alternative to the control group but not to other alternatives. Hence, we corrected this paragraph in the manuscript: “The emotional guilt and specific cost message frames showed the greatest nominal differences in no-show rates and advanced cancellation rates compared with the control group (14.2% and 15.3% compared to 21.1% in no-show rates and 26.3% and 27.4% compared to 17.2% in advanced cancellation rates, respectively)”.

• Text line 279 - “substancial“ is misspelled

Response: Thank you. We have actually rewritten this section, and the word no longer appears

• Text lines 278-281 - is your effect due to a “mere change in narrative” or specific “behavioral economics principles”? Your analyses don’t allow you to distinguish. The messages included in Supplement 1 would appear to a prudent layperson to match the principles listed in each row. But you’ve reported no independent message testing to confirm that the messages are perceived that way. This does not allow you to defend against any number of alternative explanations that have nothing to do with the specific behavioral economics principles. For example, is the difference because the effective messages have the longest word/character counts? Is this a Hawthorne effect, given that the control message had been in use for a while and most of the experimental messages were noticeable changes from the prior one? Is it because the effective messages provide a reason, any reason, to show up or cancel in advance (see Ellen Langer’s 1978 study on “placebic” information in persuasion)?

Response: We thank the reviewer for this important comment. We agree that the results only imply that different messages influence and optimize appointment behaviors even if no underlying theory is applied. It can be only speculated that the varied appointment effects are consistent with the source behavior theories. The current data do not offer adequate support that any of these behavioral theories are responsible for study effects, nor that one theory predicts stronger effects than another. We added a sentence in the discussion section (at the end of the second paragraph): 

"These results indicate that different messages can influence no-show and cancellation rates. These results are aligned with behavioral economic theories. However, the current data do not offer adequate support that the varied effects have resulted from those specific psychological mechanisms. Future research is needed to explore this topic”. 

• Text lines 285-286 - You’ve shown no data to confirm whether patients receiving these messages felt any ‘affect’ or any specific emotional response, so the current version of your study cannot be evaluated against specific frameworks such as MINDSPACE.

Response: We agree with the reviewer and have adjusted the paragraph in the discussion section accordingly.

• Text lines 306-308 - This is a minor point given the larger context of the study, but the multiple and varying number of appointments per patient in the sample leads to correlated error variance in your study dataset, which may have produced biased error estimates. If a study dataset allows some patients to count once, but others to count multiple times, a statistical analysis model that accounts for this correlation in error variances is often used. In your resubmission, it’s probably better to mention this as a limitation than to go back and apply a more sophisticated analysis model (e.g., genereralized estimating equations) that may not improve your precision.

Response: We thank the reviewer for this insightful comment. We agree that 

the multiple and varying number of appointments per patient in the sample may have led to correlated error variance in our study dataset. 

Given that the eligible population for this study included patients who had an appointment between December 2018 and March 2019, there were not many patients who had repeated appointments. Therefore, the statistical analysis model employed was not generalized estimating equations. We note that this is a potential limitation and addressed this issue in the limitations section as follows: “The multiple and varying number of appointments per patient in the sample led to correlated error variance in the study dataset, which may have produced biased error estimates. However, due to the short period of the study (December 2018 – March 2019), not many patients had repeated appointments (approximately 9.4% of the total population).”

• Text lines 327-328 - it might make sense to acknowledge that a change like this might have additional unintended consequences besides improving visit attendance. For example, does using a guilt-based message encourage patient resentment that might reduce clinic or physician satisfaction ratings?

Response: We thank the reviewer for this insightful comment, and have addressed this in the discussion section: “It is worth noting that such a change may have additional unintended consequences other than improving visit attendance, such as a reduction in clinic or physician satisfaction ratings”. 

• Text line 332 - “customize” instead of “costumize”?

Response: Yes, we meant “customize”. We have corrected this mistake.

---

## [Editor Report · Decision Letter 1]

3 Jun 2020

It’s how you say it: Systematic A/B testing of digital messaging cut hospital no-show rates

PONE-D-20-04898R1

Dear Dr. Berliner Senderey,

We’re pleased to inform you that your manuscript has been judged scientifically suitable for publication and will be formally accepted for publication once it meets all outstanding technical requirements.

Kind regards,

Sreeram V. Ramagopalan

Academic Editor

PLOS ONE
---

## [Editor Report · Acceptance letter]

8 Jun 2020

PONE-D-20-04898R1 

It’s how you say it: Systematic A/B testing of digital messaging cut hospital no-show rates 

Dear Dr. Berliner Senderey:

I'm pleased to inform you that your manuscript has been deemed suitable for publication in PLOS ONE. Congratulations! Your manuscript is now with our production department. 

Kind regards, 

on behalf of

Dr. Sreeram V. Ramagopalan 

Academic Editor

PLOS ONE